# Almond [*Prunus dulcis* (Mill.) DA Webb] Processing Residual Hull as a New Source of Bioactive Compounds: Phytochemical Composition, Radical Scavenging and Antimicrobial Activities of Extracts from Italian Cultivars (‘Tuono’, ‘Pizzuta’, ‘Romana’)

**DOI:** 10.3390/molecules28020605

**Published:** 2023-01-06

**Authors:** Simona Fabroni, Angela Trovato, Gabriele Ballistreri, Susanna Aurora Tortorelli, Paola Foti, Flora Valeria Romeo, Paolo Rapisarda

**Affiliations:** Council for Agricultural Research and Economics (CREA)—Research Center for Olive, Fruit and Citrus Crops, Corso Savoia 190, 95024 Acireale, Italy

**Keywords:** almond [*Prunus dulcis* (Mill.) DA Webb] hull, extract, triterpenoid acids, hydroxycinnamic acids, antioxidant activity, antimicrobial activity

## Abstract

In this study we developed a new extract, by the use of conventional solid-solvent extraction and a food-grade hydroalcoholic solvent, rich in phenolic and triterpenoid components from almon hull to be employed as functional ingredient in food, pharma and cosmetic sectors. Two autochthonous Sicilian cultivars (‘Pizzuta’ and ‘Romana’) and an Apulian modern cultivar (‘Tuono’) have been tested for the production of the extract. Results showed that the two Sicilian varieties, and in particular the ‘Romana’ one, present the best characteristics to obtain extracts rich in triterpenoids and hydroxycinnamic acids, useful for the production of nutraceutical supplements. About triterpenoids, the performance of the hydroalcoholic extraction process allowed to never go below 46% of recovery for ‘Pizzuta’ samples, with significantly higher percentages of recovery for ‘Tuono’ and ‘Romana’ extracts (62.61% and 73.13%, respectively) while hydroxycinnamic acids were recovered at higher recovery rate (84%, 89% and 88% for ‘Pizzuta’, ‘Romana’ and ‘Tuono’ extracts, respectively). Invitro antioxidant and antimicrobial activities exerted by the extracts showed promising results with *P. aeruginosa* being the most affected strain, inhibited up to the 1/8 dilution with ‘Romana’ extract. All the three tested extracts exerted an antimicrobial action up to 1/4 dilutions but ‘Romana’ and ‘Pizzuta’ extracts always showed the greatest efficacy.

## 1. Introduction

Almond [*Prunus dulcis* (Mill.) DA Webb] is one of the most popular nut crop all over the world. Worldwide production of almonds (in shell) is estimated at 4,140,043 tonnes in 2020 with America (including Argentina, Brazil, Chile, Mexico, Peru and USA) being the leading producer accounting to more than 50% of the global production per year. Asia (648,111 tonnes) and Europe (561,453 tonnes) follow with an estimated production which covers, for each continent, almost the 15% of the worldwide production [1] per year. Almond production in Italy is estimated at almost 85,000 tonnes per year on a cultivated area accounting to more than 50,000 ha [2]. The production is concentrated in Sicily (60%) and Apulia (30%) with negligible productions in other regions of southern Italy (Calabria, Basilicata and Sardinia). In Sicily the cultivation of the almond tree is developed in the southern area and in the south-eastern part of the region; among the native varieties, ‘Pizzuta’ and ‘Romana’ prevail. In recent years, the market demand for the consumption of almonds as dried fruit and as an ingredient in food products has significantly increased [3]. The almonds processing produces a significant quantity of waste and byproducts, which entail considerable difficulties in disposal, with the main residue being represented by the green hull which constitutes about 50% of the weight of the fresh fruit. Currently the almond by-products are mostly exploited for energetic purpose or in the livestock sector [4]. The green hull has been recently employed mostly in animal husbandry for testing its use in lamb diet [5], in broiler diet [6], on lactating cows [7] and on laying hens [8]. Beyond its use in animal husbandry, the use of almond hull for bioenergy and biofuels purposes has been also proposed with encouraging results [9,10]. Furthermore, very recently, an innovative synergistic hydrothermal co-valorization approach for biofuels production through coprocessing of almond hulls and FFP2 (Filtering Face Piece 2) disposable face masks in seawater has been proposed [11]. All these applications, even if very promising, might represent only a partial solution to the huge environmental impact of the residues of the almond dehulling process, by the way generating only low value further incomes for growers and producers. Based on its high content in bioactive compounds, including triterpenoids, lactones, phenolic compounds and fibres [12,13,14,15,16], a profitable alternative to the disposal of the raw almond hull is represented by its use in the food and pharma industries, which are high added value supply chains [17,18]. Most of the research has been focused on the chemical characterization as well as bioactivity of extracts from almond hulls. Takeoka and Dao [13] have characterized a methanolic extract of almond hull by HPLC-PDA. Furthermore, the extract was tested for its ability to inhibit the oxidation of methyl linoleate at 40 °C, and the result showed that, at the same concentration, it had a greater antioxidant activity than α-tocopherol. Studies carried out on almond hull have led to the identification of triterpenoid acids, such as betulinic, oleanolic and ursolic acids [12]. There is a growing interest in natural triterpenoids, mostly related to their broad spectrum of biological activities; they are, in fact, bactericidal, fungicidal, antiviral, cytotoxic, analgesic, antitumor, spermicidal, cardiovascular, antiallergic, anti-inflammatory and antidiabetic agents [19,20,21,22,23,24,25,26,27,28,29,30,31,32,33,34]. Moreover, ursolic acid exerts hepatoprotective activity and induction of cell apoptosis [35,36] as well as other pentacyclic triterpenoids such as betulinic acid, imberbic acid, oleanolic acid and zeylasteral have been reported to possess antimicrobial activity [37,38,39], but there are only few reports regarding the antibacterial mechanism of triterpenoids. Ursolic acid showed a synergistic effect with ampicillin and tetracycline against several bacterial pathogens [37], this makes almonds a promising source of natural antibacterial compounds, despite the mechanisms of triterpenoids action has not been clarified.

The present study was aimed at developing a new extract, rich in phenolic and triterpenoid components to be employed as functional ingredient in food, pharma and cosmetic sectors. The obtained extracts were characterized with respect to their triterpenoids and hydroxycinnamic acids quali-quantitative profile and further antiradical and antimicrobial activities were tested and validated with promising results.

## 2. Results and Discussion

### 2.1. HPLC-PDA Analysis of Triterpenoids

In Table 1 the results of the HPLC-PDA analysis of triterpenoids in ‘Pizzuta’, ‘Romana’ and ‘Tuono’ hull raw samples (H) and extracts (E) are reported.

Total triterpenoids in H samples ranged between 412.67 ± 0.27 mg/100 g DW in ‘Tuono’ samples to 457.59 ± 2.78 mg/100 g DW in ‘Pizzuta’ samples with average and not statistically different values for ‘Romana’ variety (446.95 ± 10.88 mg/100 g DW). The quantitative analysis of the three varieties showed that ‘Romana’ and ‘Pizzuta’, the two autochthonous Sicilian varieties, present higher concentrations of total triterpenoids with a predominance of oleanolic acid for ‘Romana’ variety and a predominance of ursolic acid for ‘Pizzuta’. Looking at the qualitative composition of the individual compounds, the triterpenoid profile was different in each of the three variety. Indeed, ‘Pizzuta’ hull samples showed a higher presence of ursolic and betulinic acids (162.83 ± 1.61 mg/100 g DW and 160.94 ± 1.04 mg/100 g DW, respectively) with regard to oleanolic acid (133.82 ± 0.13 mg/100 g DW). Conversely, ‘Romana’ and ‘Tuono’ hull samples showed higher levels of oleanolic acid respect to betulinic and ursolic acids. In general, statistical analysis of the acquired data showed that betulinic and ursolic acids were significantly higher in ‘Pizzuta’ H samples while lower values were recorded for ‘Tuono’ samples and average levels in ‘Romana’ samples. On the other hand, oleanolic acid was most represented in ‘Tuono’ and ‘Romana’ hull samples respect to the other variety.

The quali-quantitative triterpenoid profile in E samples of the three varieties showed higher values of oleanolic acid in ‘Romana’ samples while betulinic acid was mostly represented in ‘Pizzuta’ samples and ursolic acid was equally represented in ‘Pizzuta’ and ‘Romana’ samples respect to ‘Tuono’ samples where lower values were recorded for this compound (41.69 ± 0.34 mg/100 mL). In general, total triterpenoids acids resulted higher in ‘Romana’ samples, showing the highest % of recovery of the bioactive triterpenoids respect to the hull raw triterpenoid content (73.13%). Results showed that, on average, the performance of the hydroalcoholic extraction process allowed to never go below 46% of recovery for ‘Pizzuta’ samples, with significantly higher percentages of recovery for ‘Tuono’ and ‘Romana’ extracts (62.61% and 73.13%, respectively). Moreover, quantitatively, ‘Romana’ extracts resulted the richest in triterpenoid acids with an average value equal to 326.86 ± 6.34 mg/100 mL. It can be concluded that the results of this study show that the two Sicilian varieties, and in particular the ‘Romana’ one, present the best characteristics to obtain extracts rich in triterpenoids, useful for the production of nutraceutical supplements.

As far as the recovery of terpenoids from almond residual hull is concerned, the first study which led to the identification of triterpenoid acids, such as betulinic, oleanolic and ursolic acids, from almond hull was that of Takeoka et al. [12]. In this study, the authors report an estimated overall amount equal to ~1% of the hulls for the three triterpenoids comprised together, but no reference to the cultivar or variety of origin of the investigated hull samples is reported. With their results, the authors claimed that almond hulls could be considered as a rich source of these promising anti-inflammatory, anti-HIV and anticancer agents. Further research was published on antiproliferative terpenoids from almond hulls [40]. Our results show that hydroalcoholic extraction could be proposed as a valuable and efficient process to quantitatively recover triterpenoids from raw almond hull, especially for the ‘Romana’ cultivar which showed the most promising % of recovery associated with a valuable quali-quantitative triterpenoid profile.

### 2.2. HPLC-PDA Analysis of Hydroxycinnamic Acids

In Table 2 the results of the HPLC-PDA analysis of hydroxycinnamic acids in ‘Pizzuta’, ‘Romana’ and ‘Tuono’ hull raw samples (H) and extracts (E) are reported.

Total hydroxycinnamic acids in almond H samples showed relevant values ranging from 138.39 ± 9.42 mg/100 g DW to 303.22 ± 21.87 mg/100 g DW, respectively, for ‘Tuono’ and ‘Pizzuta’ varieties, showing that ‘Pizzuta’ variety resulted the richest in total hydroxycinnamic acids. ‘Romana’ hull samples showed average values (180.96 ± 9.63 mg/100 g DW) which did not produce any statistical difference with regard to the other two varieties. As observed for total triterpenoid acids, the two autochthonous Sicilian cultivars, with particular reference to ‘Pizzuta’, showed quantitatively higher values of total hydroxycinnamic acids respect to the ‘Tuono’ variety, confirming the high potential of the almond hull deriving from the dehulling process of these varieties, as a functional ingredient. The predominant compounds were represented by chlorogenic, neochlorogenic and cryptochlorogenic acids whose levels showed a similar trend in all the samples from the three varieties. Indeed, chlorogenic acid resulted to be the predominant hydroxycinnamic compound in all of the three varieties with values which ranged from 12.51 ± 0.18 mg/100 g DW to 122.03 ± 17.69 mg/100 g DW for ‘Tuono’ and ‘Pizzuta’ cultivars, respectively. Cryptochlorogenic showed to be the second most present compound in ‘Pizzuta’ (6.56 ± 0.12 mg/100 g DW) and ‘Tuono’ (5.85 ± 0.32 mg/100 g DW) samples with the exception of the ‘Romana’ variety in which neochlorogenic acid quantitatively followed the most abundant chlorogenic acid and preceded cryptochlorogenic acid.

The quali-quantitative profile of hydroxycinnamic acids in E samples reflected the same trend recorded for the H samples of the three investigated varieties. ‘Pizzuta’ extract was the richest in total hydroxycinnamic acids with 254.70 ± 18.37 mg/100 mL, recording a high recovery rate equal to 84% with respect to the content of the almond H samples. The ‘Romana’ extract, the other autochthonous Sicilian cultivar, quantitatively followed with an average content equal to 161.05 ± 8.57 mg/100 mL of total hydroxycinnamic acids, showing an 89% recovery rate, considering the original content of the H samples deriving from this cultivar. The ‘Tuono’ variety showed not significantly lower values (121.78 ± 8.29 mg/100 mL) with respect to the ‘Romana’ variety, with an average recovery rate equal to 88%. As observed in almond H samples, chlorogenic, neochlorogenic and cryptochlorogenic acids resulted to be the most represented compounds with chlorogenic acid showing higher values in all of the three varieties. Chlorogenic acid showed values ranging from 111.05 ± 16.10 mg/100 mL to 11.64 ± 0.17 mg/100 mL, for ‘Pizzuta’ and ‘Tuono’, respectively, with ‘Romana’ showing an intermediate concentration equal to 63.12 ± 0.59 mg/100 mL. Cryptochlorogenic acid showed higher values in ‘Romana’ hull extract with a concentration equal to 9.36 ± 0.26 mg/100 mL and lower and comparable values in ‘Pizzuta’ (5.71 ± 0.10 mg/100 mL) and ‘Tuono’ (5.15 ± 0.28 mg/100 mL) hull extracts. ‘Romana’ hull extract showed higher concentration of neochlorogenic acid (11.68 ± 0.13 mg/100 mL), if compared to the other two varieties (1.57 ± 0.02 mg/100 mL for ‘Pizzuta’ and 1.61 ± 0.05 mg/100 mL for ‘Tuono’ hull extracts, respectively). As noted in H samples, ‘Pizzuta’ and ‘Tuono’ hull extracts showed the following trend: chlorogenic acid > cryptochlorogenic acid > neochlorogenic acid while in ‘Romana’ hull extract neochlorogenic acid quantitatively followed the most abundant chlorogenic acid and preceded cryptochlorogenic acid.

The previous literature has shown the great potential of almond hull extracts as valuable bioactive ingredients in food, feed or nutra-pharma formulations. Indeed, almond hull is a rich source of terpenoids and phenolics. Takeoka and Dao, in 2003 [13], analysed by HPLC-PDA a methanolic extract of almond hull belonging to ‘Nonpareil’ variety finding that chlorogenic acid was the main phenolic compound in almond hull while cryptochlogenic was the second most abundant, followed by neochlorogenic. Our results confirm this finding showing that chlorogenic acid is the most represented phenolic compound in all of the three Italian investigated cultivars. The quali-quantitative profile of phenolic compounds in almond hull may be also affected by the cultivar, by way of the genotype that may influence the antioxidants profile, as previously reported by Sfahaln et al. [41]. In 2006, Wijeratne et al. [42] demonstrated that a green almond hull ethanolic extract (variety not reported) performed better than almond skin and seed extracts in inhibiting the formation of both primary and secondary oxidation products. In the same study, the authors analysed almond hull extract performing a HPLC-PDA analysis on free phenolic acids and those liberated from soluble esters and glycosides. Our results are also in line with this finding confirming that the main phenolic compounds in almond hull are cinnamic acids derivatives. Esfahlan et al. [43] successively reported a comprehensive review on antioxidants from almond and its byproducts, showing that hydroxybenzoic acids and aldehydes (with the only exception of protocatechuic acid reported in traces by Wijeratne et al. [44]), anthocyanidins, procyanidins and flavanone glycosides have not ever been reported in almond hull. Conversely, the authors reported that the main phenolic compounds in almond hull are represented by hydroxycinnamic acids, specifically the most represented being caffeoylquinic derivatives. The previous available literature reports traces of kaempferol-3-O-rutinoside and isorhamnetin-3-O-glucoside, as flavonol glycosides and quercetin, quercitrin and isorhamnetin as flavonol aglycons and morin as flavanone aglycone in almond hull [44]. More recently, Kahlaoui et al. [45] characterized by HPLC-PDA the polyphenolic compounds extracted by a hydroalcoholic solvent from different varieties of almond hulls, including the Sicilian ‘Pizzuta’ and ‘Romana’ ones, by comparing ultrasound assisted extraction (UAE) with conventional solid-solvent extraction (CSE). Results of this study have confirmed that chlorogenic acid is the most abundant phenolic acid in almond hull with the ‘Pizzuta’ variety being the richest. This study also highlighted that other minor phenolic compounds (catechin, protocatechuic acid, quercetin-3-glucoside, *p*-coumaric acid and epicatechin) have been identified in trace amounts only in some varieties and only in UAE, being not detectable in CSE. The output of our research, performed on hydroalcoholic extracts obtained by conventional solid-solvent extraction, confirm this finding.

### 2.3. Antioxidant Activity Assays

#### 2.3.1. Folin–Ciocalteu Reagent Assay

The results of the Folin–Ciocalteu Reagent (FCR) colorimetric assay of the hull extract of the ‘Pizzuta’, ‘Romana’ and ‘Tuono’ varities are shown in Table 3. Results showed that the total phenolic content of the Sicilian varieties (‘Pizzuta’ and ‘Romana’) is significantly higher compared to the Apulian one (‘Tuono’). Indeed, ‘Pizzuta’ hull extract accounted for 22,593.33 ± 187.53 mg GAE/100 mL while ‘Romana’ variety recorded a total phenolic content equal to 21,284.76 ± 195.24 mg GAE/100 mL, highlighting the high antioxidant value of the hull as the main almond processing byproduct. ‘Tuono’ variety showed a significantly lower total phenolic content (18,307.26 ± 94.15 mg GAE/100 mL) if compared with the two Sicilian autochthonous varieties; however, recording a relevant total phenolic content. Correlation analysis has shown a significant direct correlation between total phenolic content and chlorogenic acid (r = 0.98; *p* ≤ 0.05), betulinic acid (r = 0.91; *p* ≤ 0.05) and ursolic acid (r = 0.98; *p* ≤ 0.05) contents. One of the first paper reporting total phenolic content of an ethanolic extract of almond hull (variety not shown) is the work by Siriwardhana et al. [46], where the authors, in addition to determining the total phenolic content of almond seed, skin and hull extracts, examined the free radical-scavenging activity of almond extract by using different in vitro assays. Later on, Wijeratne et al. [39] evaluated the antioxidant activity of almond hull extract while in a contemporary work by the same authors [44] demonstrated that the total phenolic content of almond hull ethanolic extract was nine times higher than that of the whole seed. In 2009, Sfahlan et al. [41] investigated on the total phenolic content and related antiradical activity of almond (*Amygdalus communis* L.) hulls and shells methanolic extracts belonging to 18 different genotypes. The authors found significant differences in phenolic contents and radical scavenging capacities among the different genotypes. More recently, Kahlaoui et al. [45] showed the total phenolic content values of almond hull ethanolic extracts from different varieties, including the Sicilian ‘Pizzuta’ and ‘Romana’ ones, obtained by UAE and CSE. The authors showed that hull extracts from ‘Pizzuta’ displayed the highest content of polyphenolic compounds with concentrations which are comparable with those reported in our study. Qureshi et al. [47] recently reported a concentration of 1% *w*/*w* of polyphenolic compounds of a dried 70% ethanolic hull extract from almonds collected in China. Taking into consideration data from the previous literature, our results highlight the high added value of hulls from Italian almond varieties, particularly those autochthonous from Sicily, and put in evidence that a conventional solid extraction process, by using hydroalcoholic food grade solvents, can be conveniently used to recover relevant amount of polyphenolic bioactive compounds.

#### 2.3.2. ORAC Assay

Differently from FCR assay which is based on an electron-transfer (ET) reaction, the ORAC assay is based on a hydrogen atom transfer (HAT) reaction mechanism and measures the antioxidant scavenging activity against peroxyl radicals induced by AAPH radical. Therefore, it is manly correlated with those compounds of the extract which exert their antioxidant activity thanks to the transfer of a hydrogen. So, it can be said that ORAC units measure antioxidant capacity of a sample. The higher the ORAC units, the higher the antioxidant potential of the sample. Our results (Table 3) showed that the Sicilian varieties present a definitely higher antioxidant activity compared to the Apulian one. The two Sicilian autochthonous varieties showed significantly greater ORAC units (44.424 ± 1376.37 µmoles Trolox equiv./100 mL and 41.966 ± 1596.59 µmoles Trolox equiv./100 mL for ‘Pizzuta’ and ‘Romana’ almond hull extracts, respectively) respect to the ‘Tuono’ almond hull extract which accounted to 29.250 ± 591.76 µmoles Trolox equiv./100 mL. Correlation analysis has shown a significant direct correlation between ORAC units and chlorogenic acid (r = 0.94; *p* ≤ 0.05), betulinic acid (r = 0.84; *p* ≤ 0.05) and ursolic acid (r = 1.00; *p* ≤ 0.05) contents. Previous studies aimed at pointing out the antioxidant activity of almond hull extracts have revealed that it is active in (i) inhibiting the formation of both primary and secondary oxidation products [42], (ii) inhibiting human low-density lipoprotein oxidation, DNA scission and inducing metal ion chelation [44], (iii) scavenging nitrite, hydrogen peroxide and superoxide radicals [41] and (iv) inhibiting tyrosine phosphatase-1B (PTP1B) enzyme [47]. Our results reveal that almond hull extracts provide a great and relevant intake of antioxidants, also assuring an outstanding increase in ORAC units. The high antioxidant value of the hull may be ascribed to the fact that almond hull constitutes a natural protection of the oil-rich internal almond seed from oxidation induced by sunlight in the presence of atmospheric oxygen, anyway the favourable in vitro potential of any compound or extract, the safety level and the beneficial outcomes must be supported by in vivo toxicological studies [48].

### 2.4. Antimicrobial Activity

In general, triterpenoid compounds such as betulinic, ursolic and oleanolic acids, in addition to their antiviral, antitumor or anti-inflammatory properties, have been shown to exhibit antimicrobial properties against human pathogens [49]. Although the mechanism of their antimicrobial activity has not yet been elucidated, recent studies indicate that the antibacterial action of triterpene acids is due to pleiotropic effects, destroying the integrity of bacterial membranes and inhibiting metabolic pathways and protein synthesis [50,51]. Furthermore, Oloyede et al. [52] demonstrated that antimicrobial action, particularly against *Escherichia coli*, *Pseudomonas areuginosa* and *Staphylococcus aureus*, could be linked to bacterial cell death through the generation of reactive oxygen species. In Table 4, the antimicrobial activity results of the tested extracts are shown. The inhibition halos obtained with the extracts showed a dose-dependent antimicrobial action against some pathogenic strains such as *P. aeruginosa*, *S. aureus* and *E. coli*. Regarding the *S. aures* strain, these results agree with Mandalari et al. [53] and Smeriglio et al. [54] that reported that phenolic fraction of almond skins possess significant antimicrobial activity against Gram-positive bacterial strains [55]. On the contrary, the extracts showed no inhibition against *Candida albicans* as confirmed by Foti et al. [56] with different natural extracts. These results could be due to a different hydrophilic/lipophilic ratio balance that could increase cellular uptake, enhancing antioxidant or antimicrobial activities, as suggested by Diallinas et al. [57]. In detail, *P. aeruginosa* was the most affected strain showing the largest halos and the only one inhibited up to the 1/8 dilution with ‘Romana’ extract. All the three tested extracts exerted an antimicrobial action up to 1/4 dilutions but ‘Romana’ and ‘Pizzuta’ extracts always showed the greatest efficacy.

Since, the raw extracts showed a mean value of 16% dry matter (DM) content, their MIC value expressed as DM activity was 40 mg/mL for *S. aureus*, *E. coli* and *L. innocua* and 20 mg/mL of extract (for ‘Romana’ extract) for *P. aeruginosa*. However, those MIC values expressed as total triterpenoids (Table 1) correspond to a range between 0.527 mg/mL and 0.646 mg/mL for ‘Pizzuta’ and ‘Tuono’ extracts (D2 dilution), respectively, and 0.409 mg/mL for ‘Romana’ extract (D3 dilution). In addition, other authors have shown antimicrobial activity of compounds derived from hydroxycinnamic acids. In detail, chlorogenic, neochlorogenic and cryptochlorogenic acids showed an inhibitory action against *S. aureus* with a MIC value of 5 mg/mL, exhibiting bactericidal activity and inhibiting *S. aureus* biofilm formation [58,59]. Whereas, analysing the results of the present study, the MIC expressed as total hydroxycinnamic acids (Table 2) showed values between 0.304 and 0.637 mg/mL for ‘Tuono’ and ‘Pizzuta’ extracts (D2 dilution), respectively, and 0.201 mg/mL for ‘Romana’ extract (D3 dilution). The interactions among the different phytochemicals found in the ‘Romana’, ‘Pizzuta’ and ‘Tuono’ almond extracts could synergistically affect the microorganisms’ growth by increasing the inhibitory effect.

Further studies should be performed with purified isolated fractions of the raw extracts with the aim of determining the main compounds responsible of the detected antimicrobial activity. Furthermore, in order to investigate if other extracting solvents/techniques could potentially increase their antimicrobial effect, by increasing total triterpenoids and/or total hydroxycinnamic acids contents, further investigations will be planned in the near future.

## 3. Materials and Methods

### 3.1. Chemical Reagents and Standards

Triterpenoid acids (betulinic, oleanolic and ursolic acids) and hydroxycinnamic acids (chlorogenic, cryptochlorogenic and neochlorogenic acids) were purchased from Sigma Chemical Co., (St. Louis, MO, USA). Food-grade ethanol was used for the hydroacoholic extraction. All other chemicals were of analytical grade, and the solvents used for chromatography were HPLC grade (Merck KGaA, Darmstadt, Germany).

### 3.2. Plant Material

Almond hull samples from ‘Pizzuta’ and ‘Romana’ autochthonous Sicilian cultivars were sampled at the organic farm AgribioLeone located at Noto (SR) while samples from ‘Tuono’ cultivar were sampled at the organic farm Sorelle Di Pino located at Belpasso (CT). The hull samples were collected immediately after the hulling process which almonds were subjected to after harvesting from the trees. The hull samples were taken to the laboratory and subjected to drying in a ventilated drying oven at 45 ± 1 °C in order to avoid the onset of mould proliferation or other deterioration processes. Once dried (residual humidity equal to 2 ± 0.1%), the hull samples were ground with a domestic grinder to a fine powder to further proceed with the chemical characterization of the triterpenoid and hydroxycinnamic compounds and the further hydroalcoholic extraction of the bioactive molecules.

### 3.3. Extraction by Hydroalcoholic Solvent

Taking into consideration that the final aim of the study was to obtain extracts from almond residual hull rich in bioactive molecules to be characterized and potentially suggested to be employed in food or cosmetic industries, only food-grade solvents have been used. Following preliminary lab trials performed using hydroalcoholic solutions at different *v*/*v* ratios, the best extraction efficiency and yield was obtained with the ethanol:water solution 80:20 (*v*/*v*). The extraction of the bioactive molecules was carried out by placing 10 g of hull samples with 100 mL of an ethanol:water solution 80:20 (*v*/*v*) in flasks fitted with a cap and kept stirring at 120 rpm for 24 h at a temperature of 25 °C. At the end of 24 h, the hydroalcoholic extract was filtered and set aside; the residual hull was stirred again with another 100 mL aliquot of solvent, under the same experimental conditions; after 24 h this extract was also filtered and added to the first. The combined extracts were then subjected to distillation under vacuum in a rotavapor lab system in order to remove the residual ethanol. The extracts rich in bioactive molecules were then employed for further determinations of their relative content in triterpenoids and hydroxycinnamic acids, antioxidant and antimicrobial activities. The concentrated aqueous extracts had on average a dry matter content equal to 16% (16.48% for ‘Romana’, 15.98% for ‘Pizzuta’ and 16.03% for ‘Tuono’ extracts).

### 3.4. HPLC-PDA Analysis of Triterpenoids

Triterpenoid acids have been determined in hull raw samples (H) and in aqueous extracts after the hydroalcoholic extraction phase of the hull samples and the subsequent removal of the residual ethanol (E).

The H samples were characterized by HPLC-PDA analysis respect to their native triterpenoids quali-quantitative composition, prior to the hydroalcoholic extraction phase, in order to subsequently determine the extraction yield of these compounds. In brief, 10 g of the powdered hull samples were extracted with 100 mL of methanol at ambient temperature under stirring at 120 rpm for 24 h in flasks fitted with a cap. After filtration, the residual hull powder was stirred again with another 100 mL aliquot of methanol solvent, under the same experimental conditions. Then the two methanolic extracts were combined, filtered with 0.45 μm PTFE filters and injected into the chromatographic system for the determination of triterpenoid acids.

After the hydroalcoholic extraction phase of the hull samples and the subsequent removal of the residual ethanol, 500 μL of the aqueous extracts were diluted to 25 mL with methanol, then filtered with 0.45 μm PTFE filters and injected into the chromatographic system for the determination of triterpenoid acids.

All the samples were analysed by HPLC-PDA using the method of Taralkar and Chattopadhyay [60], with some modifications. The HPLC analyses were performed using the Alliance Waters 2695 instrumentation, equipped with a PDA detector. The separation was achieved on a Luna (Phenomenex, Torrance, CA, USA) C18 analytical column (250 mm × 4.6 mm, 5 μm). The mobile phase was an acetonitrile:methanol solution (80:20, *v*:*v*) and the flow rate was set at 0.5 mL/min in isocratic mode. All analyses were performed with the column temperature maintained at 35 °C and the injection volume of each sample was 20 μL. The data were analysed at a wavelength of 210 nm. All analyses were repeated three times. Quantification of triterpenoid compounds was performed using an external standard calibration curve and expressed as mg/100 g of dried almond hull or mg/100 mL of concentrated aqueous extract.

### 3.5. HPLC-PDA Analysis of Hydroxycinnamic Acids

Hydroxycinnamic acids have been determined in hull raw samples (H) and in aqueous extracts after the hydroalcoholic extraction phase of the hull samples and the subsequent removal of the residual ethanol (E).

The H samples were characterized by HPLC-PDA analysis respect to their native hydroxycinnamic quali-quantitative composition, prior to the hydroalcoholic extraction phase, in order to subsequently determine the extraction yield of these compounds. In brief, 10 g of the powdered hull samples were extracted with 100 mL of methanol (containing 0.5% of hydrochloric acid) at ambient temperature under stirring at 120 rpm for 24 h in flasks fitted with a cap. After filtration, the residual hull powder was stirred again with another 100 mL aliquot of the same methanol solvent, under the same experimental conditions. Then the two methanolic extracts were combined, filtered with 0.45 μm PTFE filters and injected into the chromatographic system for the determination of hydroxycinnamic acids.

After the hydroalcoholic extraction phase of the powdered hull samples and the subsequent removal of the residual ethanol, 500 μL of the aqueous extracts were diluted to 25 mL with methanol (containing 0.5% of hydrochloric acid), then filtered with 0.45 μm PTFE filters and injected into the chromatographic system for the determination of hydroxycinnamic acids.

All the samples were analysed by HPLC-PDA using the same instrument described above. The separation was achieved on a Luna (Phenomenex, Torrance, CA, USA) C18 analytical column (250 mm × 4.6 mm, 5 μm). The eluents used were solvent A: water:formic acid (99.7:0.3 *v*/*v*) and solvent B: acetonitrile:formic acid (99.7:0.3 *v*/*v*) with a gradient transition as follows: (0 min) 95% solvent A; (50 min) 72% solvent A; (57 min) 57% solvent A; (75 min) 57% solvent A; (80 min) 95% solvent A; (90 min) 95% solvent A. Flow rate was set at 1 mL/min and all analyses were performed with the column temperature maintained at 35 °C and the injection volume of each sample was 20 μL. The data were analysed at a wavelength of 320 nm. All analyses were repeated three times. Quantification of the hydroxycinnamic compounds was performed using an external standard calibration curve and expressed as mg/100 g of dried almond hull or mg/100 mL of concentrated aqueous extract.

### 3.6. Antioxidant Activity Assays

There are several in vitro assays to determine the antioxidant activity including ORAC, TEAC (or ABTS), DPPH, FRAP and FCR. In order to evaluate the antioxidant activity of the aqueous extracts to be potentially employed in food or cosmetic industries two methods were used: the ORAC and FCR assays. The ORAC assay represents a hydrogen atom transfer (HAT) reaction mechanism and measures the antioxidant scavenging activity against peroxyl radicals induced by AAPH, while FCR is based on an electron-transfer (ET) reaction with the oxidant as an indicator of the reaction endpoint [61].

#### 3.6.1. Folin–Ciocalteu Reagent Assay

The FCR colorimetric method was applied as described by Amenta et al. [62] with some modifications. The colorimetric method [63] usually used to determine total phenolics, was applied in this work to evaluate antioxidant activity of hull extracts. After the hydroalcoholic extraction phase of the powdered hull samples and the subsequent removal of the residual ethanol, appropriately diluted concentrated aqueous extracts (1:200, *v*/*v*) samples (1 mL) were mixed with 5 mL of FCR commercial reagent (previously diluted with water 1:10 *v*/*v*) and 4 mL of a 7.5% sodium carbonate solution. The mixture was stirred for 2 h at room temperature while avoiding strong light exposure. The absorbance of the resulting blue solution was measured spectrophotometrically at 740 nm, and the concentration of total phenolics was expressed as (±) gallic acid equivalents (mg/100 mL).

#### 3.6.2. ORAC Assay

The ORAC assay was used as described by Cao et al. [64], and improved by Ou, Hampsch-Woodill and Prior [65], adapted and modified. Briefly, the measurements were performed on a Wallac 1420 Victor III 96-well plate reader (EG & Wallac, Turku, Finland) equipped with fluorescence filters (excitation 485 nm, emission 535 nm). Fluorescein (116 nM) was the target molecule for free radical attack from AAPH (153 mM) as the peroxyl radical generator. The reaction was conducted at 37 °C, at pH 7.0 with Trolox (1 μM) as the control standard and 75 mM phosphate buffer (pH 7.0) as the blank. All solutions were freshly prepared prior to analysis. After the hydroalcoholic extraction phase of the powdered hull samples and the subsequent removal of the residual ethanol, concentrated aqueous extracts were diluted with phosphate buffer (1:200, *v*/*v*) prior to analysis, and results were reported as micromoles of Trolox equivalents per 100 mL of extract.

### 3.7. Antimicrobial Activity

The inhibitory activity of the extracts was tested on five relevant potentially pathogenic strains such as: *Pseudomonas aeuroginosa* DSM 1117, *Staphylococcus aureus* DSM 30862, *Escherichia coli* DSM 1103, *Listeria innocua* DSM 20649 and *Candida albicans* DSM 1386 (Leibniz-Institute DSMZ, German collection). The media used for the test were PDA (Potato Dextrose Agar, Oxoid, UK) for Candida albicans, while MHA (Muller Hinton Agar Base, Merck, Germany) medium was used for all other strains. Before plating, microorganism inocula were standardized to a turbidity of 0.5 McFarland standard (10^6^ yeasts or 10^8^ bacteria cells/mL). In each plate containing the selective medium, 1 mL of inoculum was spatulated, allowed to dry, and then 6 mm cellulose discs soaked with four different dilutions of extracts were placed to determine their minimal inhibitory concentration (MIC). For each extract, serial twofold dilutions indexed to the base 2 starting from each raw extract [66,67] were prepared for the test. Distilled water was used as a negative control. Plates were incubated at specific temperatures for 48 h and results expressed as diameter of the inhibition halo (mm). Each plate was set up in triplicate.

### 3.8. Statistical Analysis

Data were statistically analysed by using STATISTICA 6.0 (StatSoft Italia srl, Vigonza, Padova, Italy). The statistical differences were assessed by variance analysis (ANOVA) and means partitioning was carried out by the Tukey’s HSD test. To determine the relationships between the evaluated parameters, Pearson correlation coefficients (r; *p* ≤ 0.05) were used.

## 4. Conclusions

In this study, we have proposed almond processing residual hull as a new source of bioactive compounds by the application of a conventional solid-solvent extraction associated with the use of a food-grade hydroalcoholic solvent solution in order to obtain a phytoextract rich in bioactive components. Indeed, our results have showed that a concentrated phytoextract rich in phenolic and triterpenoid components may be obtained starting from two autochthonous Sicilian cultivars (‘Pizzuta’ and ‘Romana’) and an Apulian modern cultivar (‘Tuono’). More specifically, our results have showed that hydroalcoholic extraction could be proposed as an easy and efficient process to quantitatively recover triterpenoids from raw almond hull, especially for the ‘Romana’ cultivar which showed the most promising % of recovery associated with a valuable quali-quantitative triterpenoid profile. Furthermore, the quali-quantitative profile of hydroxycinnamic acids in concentrated extracts have showed that the ‘Pizzuta’ extract was the richest in total hydroxycinnamic acids while ‘Romana’ have showed the highest recovery rate. Taking into consideration previous works aimed at recovering polyphenolic compounds or, alternatively, triterpenoid compounds from almond hull by the use of conventional or assisted extraction methods, our results allow to demonstrate that high recovery rates for both triterpenoids and polyphenolic compounds could be achieved in a single extract by the use of a simple and not expensive hydroalcoholic solvent extraction process. Due to the presence of the hydrophobic triterpenoid backbone, triterpenoids compounds, even if generally water soluble, could exert multiple hydrophobic interactions and this is the reason why previous works have been targeted on their selective recovery without taking into consideration the possibility to simultaneously recover polyphenolic compounds. To the best of our knowledge, for the first time our work have showed that the proposed extraction method allow to obtain a combined extract rich in both triterpenoids and hydroxycinnamic acids which are recovered at high recovery rates. Further in vitro antioxidant assays have confirmed that almond hull extracts derived from the three investigated cultivars may be used as food supplement providing a great and relevant intake of antioxidants, also assuring an outstanding increase in ORAC units. Finally, the in vitro antimicrobial activity of the extracts was evaluated, and results highlighted the great potential of the new phytoextract to be employed as antimicrobic agent for different applications. *P. aeruginosa* was the most affected strain and ‘Romana’ and ‘Pizzuta’ extracts showed the greatest antimicrobial efficacy.

In conclusion, our results showed the great potential of almond green hull to be used for the recovery of valuable bioactive compounds with a wide range of applications in the food, pharma and cosmetic sectors. Further studies should be addressed aiming at validating the use of the new phytoextract as ingredient in innovative food or nutraceutical formulations.

## Figures and Tables

**Table 1 molecules-28-00605-t001:** HPLC-PDA analysis of triterpenoids in ‘Pizzuta’, ‘Romana’ and ‘Tuono’ hull raw samples (H) and extracts (E).

	Hull Raw Samples *				Extracts **				
	Oleanolic Acid (mg/100 g DW)	Betulinic Acid (mg/100 g DW)	Ursolic Acid (mg/100 g DW)	Total Triterpenoids (mg/100 g DW)	Oleanolic Acid (mg/100 mL)	Betulinic Acid (mg/100 mL)	Ursolic Acid (mg/100 mL)	Total Triterpenoids (mg/100 mL)	Recovery (%)
‘Pizzuta’	133.82 ± 0.13 ^B^	160.94 ± 1.04 ^A^	162.83 ± 1.61 ^A^	457.59 ± 2.78 ^a^	60.26 ± 2.09 ^C^	75.29 ± 1.99 ^A^	75.41 ± 1.11 ^A^	210.96 ± 5.02 ^C^	46.11
‘Romana’	276.03 ± 5.67 ^A^	73.97 ± 1.78 ^B^	96.95 ± 3.44 ^B^	446.95 ± 10.88 ^ab^	204.88 ± 3.29 ^A^	50.27 ± 0.96 ^B^	71.72 ± 2.21 ^A^	326.86 ± 6.34 ^A^	73.13
‘Tuono’	285.71 ± 0.19 ^A^	57.98 ± 0.05 ^C^	68.97 ± 0.12 ^C^	412.67 ± 0.27 ^b^	178.28 ± 3.41 ^B^	38.39 ± 1.04 ^C^	41.69 ± 0.34 ^B^	258.37 ± 4.31 ^B^	62.61

Data are expressed as means of three analytical replicates ± standard deviation; Means in the same column followed by different letters are significantly different: *p* ≤ 0.01—capital letters; *p* ≤ 0.05—small letters. *: abbreviated in the text as H samples; **: abbreviated in the text as E samples.

**Table 2 molecules-28-00605-t002:** HPLC-PDA analysis of hydroxycinnamic acids in ‘Pizzuta’, ‘Romana’ and ‘Tuono’ hull raw samples (H) and extracts (E).

	Hull Raw Samples *				Extracts **				
	Neochlorogenic Acid (mg/100 g DW)	Chlorogenic Acid (mg/100 g DW)	Cryptochlorogenic Acid (mg/100 g DW)	Total Hydroxycinnamic Acids (mg/100 g DW)	Neochlorogenic Acid (mg/100 mL)	Chlorogenic Acid (mg/100 mL)	Cryptochlorogenic Acid (mg/100 mL)	Total Hydroxycinnamic Acids (mg/100 mL)	Recovery (%)
‘Pizzuta’	1.85 ± 0.03 ^B^	122.03 ± 17.69 ^A^	6.56 ± 0.12 ^B^	303.22 ± 21.87 ^A^	1.57 ± 0.02 ^B^	111.05 ± 16.10 ^A^	5.71 ± 0.10 ^B^	254.70 ± 18.37 ^A^	84
‘Romana’	13.28 ± 0.15 ^A^	70.92 ± 0.66 ^AB^	10.34 ± 0.28 ^A^	180.96 ± 9.63 ^AB^	11.68 ± 0.13 ^A^	63.12 ± 0.59 ^AB^	9.36 ± 0.26 ^A^	161.05 ± 8.57 ^B^	89
‘Tuono’	1.75 ± 0.06 ^B^	12.51 ± 0.19 ^B^	5.85 ± 0.32 ^B^	138.39 ± 9.42 ^B^	1.61 ± 0.05 ^B^	11.64 ± 0.17 ^B^	5.15 ± 0.28 ^B^	121.78 ± 8.29 ^B^	88

Data are expressed as means of three analytical replicates ± standard deviation; Means in the same column followed by different letters are significantly different: *p* ≤ 0.01—capital letters. *: abbreviated in the text as H samples; **: abbreviated in the text as E samples.

**Table 3 molecules-28-00605-t003:** Total Phenolic Content (TPC) and Oxygen Radical Absorbance Capacity (ORAC) units in ‘Pizzuta’, ‘Romana’ and ‘Tuono’ hull extracts.

	TPC (mg GAE/100 mL)	ORAC Units (µmoles Trolox Equiv./100 mL)
‘Pizzuta’	22,593.33 ± 187.53 ^A^	44,424 ± 1376.37 ^A^
‘Romana’	21,284.76 ± 195.24 ^A^	41,966 ± 1596.59 ^A^
‘Tuono’	18,307.26 ± 94.15 ^B^	29,250 ± 591.76 ^B^

Data are expressed as means of three analytical replicates ± standard deviation; Means in the same column followed by different letters are significantly different: *p* ≤ 0.01—capital letters.

**Table 4 molecules-28-00605-t004:** Results of antimicrobial activity assays of the extracts of the Romana (R), Pizzuta (P), Tuono (T) almond hulls.

Microorganism	Cultivar	RE	D1	D2	D3	D4	Control
*P. aeruginosa*	R	11	10	8	7	0	0
	P	10	10	10	0	0	0
	T	9	9	8	0	0	0
*S. aureus*	R	9	8	8	0	0	0
	P	10	7	0	0	0	0
	T	8	0	0	0	0	0
*E. coli*	R	10	8	7	0	0	0
	P	8	8	7	0	0	0
	T	7	7	0	0	0	0
*L. innocua*	R	9	9	7	0	0	0
	P	8	8	7	0	0	0
	T	6	0	0	0	0	0
*C. albicans*	R	0	0	0	0	0	0
	P	0	0	0	0	0	0
	T	0	0	0	0	0	0

Data are expressed as diameter of the inhibition halos (mm). Legend: RE = raw extract; Control = water; D1 = extract 1/2 diluted; D2 = extract 1/4 diluted; D3 = extract 1/8 diluted; D4 = extract 1/16 diluted.

## Data Availability

Data are available from the authors.

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
