# Peer review of "Almond [Prunus dulcis (Mill.) DA Webb] Processing Residual Hull as a New Source of Bioactive Compounds: Phytochemical Composition, Radical Scavenging and Antimicrobial Activities of Extracts from Italian Cultivars (‘Tuono’, ‘Pizzuta’, ‘Romana’)"

_molecules, 2023, doi:10.3390/molecules28020605_

Round 1

Reviewer 1 Report

This study analyzed the phytochemical composition and some biological activities of almond hull. The topic is interesting, however, the following are several questions and suggestions:

- Using Folin-Ciocalteu colorimetric assay, the total phenolic content was determined. There are other assays to determine the antioxidant activity including ORAC (which was used), TEAC (or ABTS), DPPH, FRAP

- Most sections, including Abstract, Introduction, Discussion, Conclusion, are too long and need to be shortened and concentrated

- All the sections should be divided into paragraphs for easier reading and understanding 

- Acronyms/Abbreviations should be defined the first time they appear in each of three sections: the abstract; the main text; the first figure or table
- Reporting the results as “mg/100 g” or “mg/100 mL” would look more professional

- In Discussion: table data should not be repeated but emphasized and the most important observations should be discussed  

- the following idea should be added in Discussion - Despite the favorable in vitro potential of any compound or extract, the safety level and the beneficial outcomes could only be determined through in vivo toxicological studies (Vedeanu et al. Doi: 10.1071/EN19249)  

- please mention the meaning of A, B, C and a, b in all tables

- Line 267: “almond”

- Lines 290, 527: “Ciocalteu” 

- Line 292: “are shown”

Reviewer 2 Report

The work is in the field of the use and valorization of by-products, specifically from the green carcass (residue) that covers the almonds.

The work is interesting and well developed, however, there are some problems or opportunities for improvement that are detailed below:

Include in the title the scientific name of the almond in parentheses.

Review the use of period or comma to separate thousands and also decimals, in some parts of the text they are used interchangeably and that can be misleading.

In I38 does it refer to America (North America including Canada?) or does it refer only to the USA?

In l39 is it important to add that it refers to quantities per year?

In l72 clarify what is meant by ffp2?

l79 check thousands/decimals

l108 It would be good to add some other references, it is understood that everything comes from reference 19, it is necessary to include and consult more articles.

In the introduction little is said about antimicrobial activity, the work would improve if some information on this matter were included.

It is very striking that the results appear first and then the section on materials and methods. It is probably a compilation error, but it is essential to structure the manuscript logically and correctly.

It is necessary to review and correctly identify the column headings.

In the tables it would be better to use superindices to express the significant differences between samples/treatments.

l295 review thousands separators and commas.

l398 Replace legenda by legend

It is necessary to improve the analysis and discussion of the microbiological part, as it is basically qualitative and does not consider quantitative aspects.

l434 when talking about a ventilated oven, does it refer to a drying oven? review

The methodological part could be improved by including uniformly the brands and models of the equipment used. It is not necessary to use quotation marks to specify names/brands of equipment.

l440 Use extracts instead of phytoextracts.

Review and do not repeat information, HPLC-PDA is very similar for triterpenoids and hydrocyanamic acids, only the differences for the last analysis could be mentioned.

The conclusions should be rewritten entirely, they seem more like a summary than what is concluded as a result of the work and that in short is the contribution to the discipline of the article.

Round 2

Reviewer 1 Report

The authors addressed the suggestions. I have no further comments.

Author Response

Many thanks for the time spent reviewing our manuscript.

Reviewer 2 Report

The authors provided a satisfactory response to all of this reviewer's observations. 

Only the headings of tables 1 and 3 need to be revised.

Author Response

many thanks for your suggestion. We have revised all column headings and improved the tables formatting. We really hope we have satisfied your request